# Lessons Learned from All for Them: Best Practices for a Cross-Collaboration Approach to HPV Vaccination in Public Schools

**DOI:** 10.3390/vaccines11050946

**Published:** 2023-05-05

**Authors:** Paula M. Cuccaro, Jihye Choi, Efrat K. Gabay, J. Michael Wilkerson, Diane Santa Maria, Sanghamitra M. Misra, Mayra Aguilar McBride, Sally W. Vernon

**Affiliations:** 1Department of Health Promotion and Behavioral Sciences, School of Public Health, The University of Texas Health Science Center at Houston, 7000 Fannin St., Houston, TX 77030, USA; 2Center for Health Promotion and Prevention Research, School of Public Health, The University of Texas Health Science Center at Houston, 7000 Fannin St., Houston, TX 77030, USA; 3Cizik School of Nursing, The University of Texas Health Science Center at Houston, 6901 Bertner Ave., Houston, TX 77030, USA; 4Baylor College of Medicine, 1 Baylor Plaza, Houston, TX 77030, USA

**Keywords:** school health, human papillomavirus, vaccination, adolescents, implementation

## Abstract

The Community Preventive Services Task Force endorses vaccination programs in schools to increase access to vaccinations. However, implementing a school-based approach requires substantial coordination, planning, and resources. All for Them (AFT) is a multilevel, multicomponent approach to increase HPV vaccination among adolescents attending public schools in medically underserved areas in Texas. AFT comprised a social marketing campaign, school-based vaccination clinics, and school nurse continuing education. Process evaluation metrics and key informant interviews to understand experiences with AFT program implementation informed lessons learned. Lessons emerged in six domains: strong champion, school-level support, tailored and cost-effective marketing approaches, mobile provider collaboration, community presence, and crisis management. Strong support at district and school levels is vital for gaining principal and school nurse buy-in. Social marketing strategies are integral to program implementation and should be adjusted to maximize their effectiveness in motivating parents to vaccinate children against HPV, which also can be achieved through increased community presence of the project team. Preparing contingency plans and flexibility within the program can facilitate appropriate responses to provider restrictions in mobile clinics or in the event of unforeseen crises. These important lessons can offer useful guidelines for the development of prospective school-based vaccination programs.

## 1. Introduction

According to the Centers for Disease Control and Prevention, human papillomavirus (HPV) is one of the most common sexually transmitted infections, affecting an estimated 79 million Americans [1]. In the United States, approximately half of all new cases of sexually transmitted infections, including HPV, occurred in adolescents and young adults 15–24 years old in 2018 [2]. Fortunately, HPV infections are vaccine preventable. HPV vaccines are known as safe, effective measures to prevent genital warts and HPV-associated cancers including cervical, vulvar, vaginal, penile, anal, and oropharyngeal cancers [3]. While the Advisory Committee on Immunization Practices (ACIP) recommends routine vaccination for adolescents 11 or 12 years of age for maximum protection against HPV-related diseases, vaccination rates continue to be suboptimal among adolescents. As of 2020, only 54.5% of adolescents ages 13 to 15 years completed the HPV vaccine series, remaining well below the 80% target goal of Healthy People 2030. These low HPV vaccination rates, which lag behind other recommended adolescent vaccines such as Tdap and MCV4, highlight the need for catch-up vaccination strategies for individuals over the age of 13, who fall outside of the primary recommended schedule [4]. Poor HPV vaccination rates among adolescents in minority and underserved populations who often experience higher rates of HPV-associated diseases [5] are of particular concern. For example, racial and ethnic minorities were 8.6% less likely than non-Hispanic whites to follow through with the full HPV vaccine series [6], and adolescents whose caregivers are foreign-born were less likely to receive the HPV vaccine than adolescents with US-born caregivers [7,8]. While reasons for such disparities need further investigation, developers of catch-up vaccination interventions should be cognizant of these disparities and consider the sociodemographic circumstances of their adolescent recipients.

HPV vaccines in the US have traditionally been delivered in clinical settings, but the transition from childhood to adolescence brings a decline in the frequency of routine visits to healthcare providers [9]. In lieu of providing clinic-based immunization, partnering with and using schools as venues for administering routinely recommended adolescent vaccines has gained prominence, as the priority populations can be easily reached and the logistics associated with the setting can ameliorate constraints for parents, such as scheduling issues [10]. Since schools are the primary contexts in which adolescents spend most of their time outside the home, no-cost vaccinations in schools are considered an effective and efficient means of ensuring high vaccine coverage of adolescents, especially for those who are medically underserved [11,12]. School-based HPV vaccination serves as a promising opportunity to create a health-promoting environment where vaccine-eligible adolescents and their parents can be educated about HPV infections and the importance of engaging in early prevention of HPV-related diseases. While very few states in the US currently have school-entry mandates for the HPV vaccine, there are states demonstrating the effectiveness of vaccine mandates. For example, Rhode Island, the state with the highest HPV vaccination rate (70.8%) in the US [13], has demonstrated success in reaching adolescents and providing in-school HPV vaccines bundled with other adolescent vaccines at no cost through their “Vaccinate Before You Graduate” initiative [14]. This program attributed its success to their transparent communications plan to increase HPV vaccination coverage, as well as accurate and timely information for providers, provider office staff, school nurses, teachers, school administrators, parents, and the public alike [14]. For states in which legislators are hesitant to mandate HPV vaccination, alternative approaches are needed to increase vaccine completion.

The purpose of this article is to provide an overview of the development and implementation of All for Them, a comprehensive, multilevel, multicomponent program to increase initiation and completion of the HPV vaccine series among middle-school-aged adolescents attending public schools in medically underserved areas in Texas, a state without an HPV vaccine mandate. The primary objective is to describe key lessons learned from implementing All for Them and to offer recommendations for prospective school-based HPV vaccination programs.

### Program Overview

All for Them (AFT) was developed for implementation in a large urban Texas public school district in 2017 and began expanding in 2020 to include additional school districts varying in geographic region, size, and population. Project stakeholders included the partner school district(s), the participating middle schools, healthcare providers (i.e., mobile clinic partners), non-profit organizations focused on cancer prevention and immunizations, communication and content experts, and parents. Each of these stakeholder types was involved at various levels of the project, from conceptualization, development, and implementation to evaluation of the intervention.

The original district partner was a decentralized independent school district, which included 30 unique participating middle schools (28–29 campuses participated each year between 2017–2020). Schools were eligible to participate if they had marked attendance boundaries in or adjacent to medically underserved areas and a majority racial and ethnic minority student population. Schools were recruited through individual informational meetings with principals and nurses and given the opportunity to commit to the project or not, depending on their capacity, and re-recruited as needed due to principal and nurse turnover. The population in the priority schools comprised 69.3% Hispanic and 28.6% black students, with 83.3% designated as economically disadvantaged [15]. As racial and ethnic minority populations in the US are disproportionately affected in several ways regarding HPV and HPV-related cancers [6], AFT provided safety-net services for students in these populations who were unable to access vaccination services easily and affordably.

AFT comprised three primary evidence-based strategies implemented at multiple levels: (1) a parent-focused social marketing campaign; (2) comprehensive school-based vaccination clinics; and (3) school nurse continuing education. The social marketing campaign used empathy and empowerment to reframe HPV vaccination as cancer prevention and normalize the vaccine as part of routine adolescent healthcare. The linguistically and culturally appropriate educational messages in the campaign were designed to increase HPV vaccine knowledge and positive attitudes for parents at different stages of readiness to adopt the HPV vaccine for their adolescent. At pre-consented school-based vaccination clinics in public schools, students were offered five ACIP-recommended adolescent vaccines at no cost from 2017 to 2020 (since Fall 2020, students have been offered all childhood and adolescent vaccines). The project team delivered consent packets to participating middle schools, which were then distributed to all enrolled students to take home to their parents. Interested parents completed and signed the forms, indicating the vaccines they consented for their child to receive. Students who returned completed forms to the school in advance received the vaccines needed on the scheduled clinic day. Parents were not required to be present during immunization. This successful strategy had a 96% parental acceptance rate for HPV vaccination among students participating in AFT clinics who needed a dose of HPV vaccine [16]. Vaccines were administered by mobile clinic partners external to the school system. While school nurses in Texas do not routinely administer vaccines to public school students, this intervention provided education to increase their knowledge, positive attitudes, and effective communication with parents regarding the HPV vaccine. The goal was for school nurses to be confident in supporting parents’ decision-making and to make a strong recommendation for HPV vaccination. The continuing nursing education (CNE) course was free for Texas school nurses and provided continuing education units. It was offered in both in-person and online formats from 2018 to 2020 in the partner school district. This educational strategy for nurses included raising awareness for and skills training to enhance vaccine data recordkeeping to improve HPV vaccine record entry and management in school data systems. Figure 1 depicts the development, implementation, and evaluation process for AFT from 2017 to 2020 across the three strategies described above.

## 2. Materials and Methods

### 2.1. Process Evaluation

A process evaluation was conducted to assess whether the program was reaching its priority audience as intended and to provide a clear description of the context in which the program was implemented. A range of metrics were recorded in a process evaluation log as part of the implementation process. The process evaluation log included the following data: the number of students enrolled, the number of students vaccinated at the clinic, the number of ineligible students referred to outside clinics, school staff turnover, the number of clinic awareness postcards mailed home, the number of consent packets delivered and returned, the number of phone calls received from parents at each school, the amount of time spent to review packets, and total clinic time. Staff observations were also translated into the process evaluation log. We documented the level of attendance and engagement at school- and community-based parent events, and whether the schools put up posters, offered incentives to students to participate, pursued extra recruitment tactics, such as calling parents of vaccine non-compliant students, and encountered any challenges with program implementation. We also documented whether the schools implemented both the social media and other communication strategies we provided, including call-outs and text messages. Among the aforementioned components, school and community-based parent events, extra recruitment tactics, and challenges the schools faced with program implementation were the three most informative and valuable components for understanding the lessons learned and how we modified our processes accordingly to improve program implementation.

### 2.2. Marketing Feedback

At the planning and development stages of the program, we hired and worked with a local marketing firm with experience in the arena of health and social causes to develop the social marketing campaign including tactics for social media campaigns. The marketing firm then carried out the implementation of paid social media advertisements in English and Spanish, carefully monitoring the reach and level of engagement of the advertisements. Throughout the program, the project team was responsive to situational factors that affected advertising performance, and adjusted social media tactics accordingly, which entailed modifying messages conveyed by the advertisements as well as changing the advertisement type to prioritize engagement. Moreover, social media tactics were scaled according to the school’s size and, in subsequent years, previous performance. During re-assessment, the marketing firm identified ways to maintain the performance of high-performing schools while adding extra tactics (both physical and digital collateral) to mid- and low-performing schools to elevate them to higher performance levels. School-based support included provision of bilingual scripted messages that schools could use for different avenues of clinic promotion. Using the schools’ existing communication systems (e.g., robocalls, text messages, social media channels) allowed the outreach to be conducted at no additional cost to the project or to the schools. Additional tactics included direct outreach by nurses to parents, nurse incentives, and a video for teachers with a project overview and instructions for supporting the nurse. These approaches were revised or eliminated if they did not turn out to be feasible to implement or as effective as anticipated.

### 2.3. Key Informant Interviews

We conducted semi-structured interviews as a quality improvement measure to generate feedback from leaders at participating schools to enhance the program as it was being implemented. Interviews were conducted with school nurses, coordinators, and principals (n = 21) who participated in AFT to understand their experiences with the program and what could be undertaken to improve the process, including increasing parent and student participation. We conducted rapid qualitative analysis to efficiently incorporate the key informants’ thoughts and suggestions as relevant evidence to the lessons learned. The interview guide included questions about the priorities of their role as a school nurse/principal, perceived support for program implementation, planning and clinic logistics, perceived ownership of the program, parent engagement, and personal beliefs about vaccination.

## 3. Results

Lessons learned from the AFT project emerged in six key domains: (1) strong champion; (2) school-level support; (3) tailored and cost-effective marketing approaches; (4) collaboration with mobile clinic organizations; (5) importance of community presence; and (6) crisis management. See Table 1 for a synopsis of the lessons within each domain.

### 3.1. Strong Champion

Collaboration and coordination with a district-level champion (in this case, the district’s Director of Health and Medical Services) since the inception of the program facilitated direct access to leadership at individual schools and effective communication to parents. With strong support for and commitment to the program from leadership at the district level, school principals were more likely to participate and remain engaged in the school-based vaccination program. Recruitment of schools, particularly in a decentralized school system where involvement in multilevel interventions is subject to the discretion of principals [17], is influenced by whether community or district-level leadership lends importance and credibility to the intervention of interest [18]. A recent study examining key stakeholder perspectives on school-based HPV vaccination programs in North and South Carolina highlighted that collaborations between school systems and community leaders in the states have supported initiatives such as having county health department nurses come into schools and provide required vaccines to increase HPV vaccine uptake in rural areas [19]. Furthermore, we made our partnership with the district and its role as a driver of the initiative transparent to parents by having the cover letter in the clinic consent packet signed by the Director of Health and Medical Services. It is likely that this communication resonated positively with parents, provided them with a sense of assurance, helped avoid an initial mistrust they might have had of an “outside” program, and possibly alleviated hesitancy towards the HPV vaccine. Turnover of superintendents and staff in public schools continues to be a persistent problem in the US [20], and our experience reflected this trend. We faced a high rate of principal, nurse, and other staff turnover in participating schools each year, which posed a challenge of having to re-recruit schools experiencing leadership turnover. Nonetheless, having a strong district-level champion allowed new principals and nurses to transition and continue implementing AFT with confidence, which was crucial to ensuring program continuity across academic years.

### 3.2. School-Level Support

Collaboration with schools and their support were integral to achieving success of the school-based vaccination program but due to working with a large, decentralized school district, recruitment and retention of schools were inevitable challenges. Consistent and well-timed communication with school administrators through a designated school liaison on our project team played a key role in maintaining school partnerships from year to year. Given the unique needs and capacity of individual schools, school-level champions were identified on an ad hoc basis by working with the principal and nurse to select individuals most appropriate to lead this effort at each campus. In some cases, where the nurse did not act as champion, school staff such as wraparound specialists, magnet coordinators, and Communities in Schools specialists filled that role. Individuals in these roles are usually in close contact with students and have trusted relationships with parents; therefore, they served as excellent program advocates and de facto champions, particularly if they exhibited personal enthusiasm for the project outcomes. Engaging with these staff facilitated access to parents and streamlined the overall rollout of the program. Implementation of a large-scale school-based vaccination program in school settings often requires staffing needs that exceed the capacity of school nurses. Nurses are not able to meet program needs alone, and recruitment of additional coordinators is therefore crucial [21]. When schools offered non-medical coordinators, they assisted school nurses with other essential responsibilities including parent communication and the distribution and collection of consent packets. This approach allowed the nurses to focus on tasks that only they could do, such as accessing and printing student vaccination records prior to the clinic events, and ultimately bolstered their participation in the school initiative. Three nurses who participated in key informant interviews had the following to say:


*I need one clerk or somebody that can hold the students or go get the students because I got a lot of other business we’re doing and it’s very hard to get that from administration. If I was going to add something, I would make it a requirement that if so many students are expected, the administration has to provide assistance to the nurse in managing the students that day. That has to be a baseline item. The project needs more than one person if you want it to be successful.*
(Nurse 9)


*Nothing stops in the school just because we have the clinic and that’s my biggest thing being pulled away. That was a tedious task to get the packets passed out in such a short amount of time…my assistant and I were a little overwhelmed. When we’re getting the packets, there was so much chaos. It was very busy here in the clinic and then we have a procedure we have to do. But we were able to, in between the procedures and the clinic, organize, get the kids down, and help with getting them immunized. I wish I had more people in the office from my school’s side to help when the clinic comes. I did appreciate that the volunteer nurse students came and we were able to look up some of the kids’ information.*
(Nurse 6)


*The principal could introduce and say, “this is from the principal, I’m supporting this”. That was very good for us, making sure that we had the parking spaces and the building available. Making sure that we had some backup to get the kids back and forth from the classes to there and back… So the principal was on board, staff was on board. It took a lot of team effort.*
(Nurse 3)

It was also beneficial to designate a bilingual district-level nurse coordinator to support all nurses in participating schools and step in for schools without a nurse. For this intervention, it was important to select a coordinator who could communicate with parents in both English and Spanish, as the priority population in the partner school district comprised nearly 70% Hispanic students, with many parents being primarily Spanish-speaking. The district nurse enabled access to protected immunization data and resources, which facilitated clinic event implementation and data monitoring activities and had direct access to parents, encouraging clinic participation for vaccine non-compliant students. She also verified and supported vaccine data entry into students’ district records and trained school nurses to record recommended vaccines and generate vaccine compliance reports. Thus, success of the school initiative is largely influenced by the support school nurses receive at the campus level.

School nurses are not authorized to enter vaccines into the state immunization registry because they are not immunization providers of record. Therefore, we encouraged nurses to routinely enter all recommended vaccines, rather than only school-required vaccines, into district records. Promoting this data entry with school nurses is instrumental in reducing overvaccination, which can increase cost-effectiveness of vaccination programs. It can also contribute to understanding HPV vaccination rates and identifying opportunities for improvement, as well as providing parents with a comprehensive vaccination record for their child. Training and technical assistance should be provided for those nurses who do not know how to enter all vaccines into their student record systems.

### 3.3. Tailored and Cost-Effective Marketing Approaches

As communication with parents and the general community is a critical component to achieving clinic success, marketing had to be carefully planned and executed. Marketing approaches should be tailored to individual schools and communities when possible, with an understanding of how different communities in the priority population are likely to be persuaded, given varying degrees of vaccine hesitancy across the population and varying determinants of hesitancy across communities [22]. It is also important to be aware that marketing and communications strategies sometimes need to be adapted depending on various circumstances of the community or school district. For example, some schools have parents active on social media, while others may prefer in-person interactions with printed materials sent home. One principal noted:


*When we initially distributed the information it was all of course via social media, things like that. But some parents, I would say the majority of our parents, really liked that one-on-one, face-to-face kind of personalization, so that they can feel free to ask their questions.*
(Principal 3)

Moreover, marketing strategies are to be modified as needed, even if it means discontinuing activities that are not working as anticipated. We replaced mobile billboard advertising with large banners outside of schools as principals recommended using “something like voting signs that will catch their eye and remind them of clinic day” (Principal 2). Nurses and principals collectively expressed that a printed banner hanging outside the school in a visible place such as the drop-off area was well received, resulting in increased parent inquiries to our team for information. We piloted mainstream radio advertisements but received phone calls about the advertisements mainly from parents outside our partner district. Thus, we switched to Spotify digital radio advertisements, which increased visits to our website dramatically from ad clicks. As most organizations face budget constraints, we advise assessing critical needs and internal skills before reaching out to a marketing firm to save money and time. We managed the components that we could internally to avoid surcharges but outsourced when needed to ensure high-quality products, letting the marketing firm conduct tasks requiring their expertise. For example, the marketing firm managed creative design and social media advertising, while we acquired and managed printed materials.

### 3.4. Mobile Vaccination Clinic Collaboration

Given that mobile clinics are an efficient means for providing health services [23], partnering with an established and trusted provider with mobile capacity can assure parents and the school district and increase the district’s willingness to implement the vaccination program. An optimal mobile clinic organization is one that is experienced in providing childhood and adolescent healthcare services, suitable for on-campus vaccination programs, flexible to refine clinic protocols based on program needs, and ideally has an existing partnership with the district. Awareness of provider restrictions and limitations is pertinent during the selection and planning phase to minimize disruption to the project. Provider restrictions may include availability on preferred dates, amount of time the provider can allocate for a clinic, capacity constraints (the number of providers and the number of people who can be vaccinated per provider), and institutional eligibility protocols. For instance, one of our provider partners routinely administered HPV vaccines only to those 11 years of age or older prior to 2019 (at which point, they began vaccinating at age 9), and all providers were unable to vaccinate students with Children’s Health Insurance Program (CHIP) or private insurance coverage, which nurses described as “the biggest barrier” to vaccine eligibility. Nurses also explained that in the future, they would need to allocate “enough time to verify what the kids need and see if they have CHIP” (Nurse 6) to avoid any misunderstanding on the day of the clinic. These restrictions, which hinder efforts to vaccinate everyone, can impact the overall effectiveness of the program [23,24,25]. During AFT implementation, to reduce the impact of these restrictions and missed opportunities, children with CHIP or private insurance were referred to private pediatric clinics and federally qualified health centers in the area, in which providers had been trained to make strong HPV vaccine recommendations.

### 3.5. Community Presence

A prominent and continuous presence of the project team in the community is essential to foster positive relationships with the school and parent community and build parents’ trust of the program. As schools are unable to release parent information, our team had to rely on the schools to communicate with parents. Limited access to parents remained a challenge in disseminating information about the program to parents throughout the project period. In the presence of such challenges, we recommend that project staff attend as many parent-focused, school, and district events as possible to interact with parents and provide them with information about the program. Informing parents of upcoming clinics, answering questions, and assisting them in paperwork completion are all important to achieving maximum clinic participation. The following are excerpts from key informant interviews with two nurses and one principal:


*I can’t sit down with every parent and spend ten minutes. I can send these pieces home but if I don’t do any follow-up, it just kind of goes in the trash. So, I think that it helped because you were able to talk one-on-one and show the importance of the vaccines to the parents. When there is personal connection that you made people are more apt to…to want to come in. That’s the ideal around fostering opportunities for parents to get more comfortable.*
(Nurse 3)


*We do have Parent Night and because I don’t feel I’m the authority for HPV to really be able to give them everything, I think it would be really great to invite you out and I can have a timeframe allotted so that you could speak to the parents and let them know. I think that would be amazing.*
(Nurse 16)


*You could set up booths or tables when we would have open house night or things like that just to communicate this is what we’re doing. I mean that time is actually perfect. The beginning of the year when they’re actually coming into the schools they could get all the paperwork and get a little teaser of “Hey, this is coming up during open house during September” and they get actual more information. And being present a little bit more so that the parents know when a van is out there… if they know they have the resources, parents are more apt to follow regulations and guidelines.*
(Principal 1)

These interactions have also been reported as opportunities to address some of the existing barriers to school-based vaccination programs, such as parents’ skepticism and questions about reviewing students’ vaccination records [26]. When in-person interactions are not possible, another option to consider is hosting or attending virtual events. This method was particularly effective during the first months of the COVID-19 pandemic when schools were not hosting in-person events. In general, efforts to work closely with the parents are key to increasing clinic participation among students and program engagement.

### 3.6. Crisis Management

In the stage of program development, we encourage planning strategies and tactics for timely response and alternative actions for unforeseen crises or competing events that may cause disruptions to program implementation. In August 2017, Hurricane Harvey led to unplanned prolonged school closures in multiple school districts in Texas and, consequently, affected AFT implementation. The kick-off of clinic events in participating schools was postponed until the schools recovered. Further, some displaced children were no longer attending participating schools. The outbreak of the COVID-19 pandemic in Spring 2020 had a profound impact on the project as it forced us to cancel activities in the final three schools of that academic year following school closures across the state, including our partner district. However, to accommodate school and community needs, we were able to reschedule two of those schools for November 2020 by changing our clinic protocol to a drive-through process to adhere to social distancing and limited in-person contact. Similarly, outdoor events, increased online activities, and stakeholder engagement through teleconferencing were strategies that allowed a safe environment for middle school students to receive vaccinations in another school-based HPV vaccination project in Texas during the pandemic [27]. As seen through such adaptations, efforts should focus on executing timely responses and flexible arrangements with strengthened preparedness so that as much of the program as possible can be carried out as planned, rather than completely suspending the program [28]. For example, our continuing nursing education course had already been converted to a web-based program and the Director of Health and Medical Services was able to encourage nurses to take the course as professional development during the early days of the pandemic, when schools initially closed. In addition, we could continue to post educational and awareness messages to social media, even though we could not advertise upcoming clinic events.

## 4. Discussion

The important lessons learned from the implementation of AFT are commensurate with the complexity of the program and prompt us to consider elements of program success and potential barriers from the perspectives of all key individuals involved. We found that securing partnership with district leaders and support from participating schools is pivotal to ensuring buy-in from principals for program implementation in their schools. As program champions, school nurses are more successful at supporting school-based vaccination with school-level support, and their engagement may also dispel HPV vaccine hesitancy among parents. While challenges inherent to working with schools remain, such as school staff turnover, lack of assistance for school nurses, and access to parents, continued efforts to bolster community awareness of the program and the importance of HPV vaccination can counter some of these challenges as schools prioritize adolescent HPV vaccination based on community demand. However, we acknowledge that school-based vaccination programs may not be feasible or suitable in certain school systems due to competing priorities, capacity for implementation, or the availability of healthcare professionals to administer vaccines. Fine-tuning social marketing strategies based on program objectives can facilitate greater community awareness and motivate parents to vaccinate their children against HPV. Finally, we recommend developing a full understanding of the capacity and limitations of selected mobile clinic providers as they are critical partners in the program. Modifications may be necessary to accommodate provider limitations to reduce missed opportunities. Similarly, flexibility and contingency plans are important, particularly in times of unforeseen emergencies that can derail program implementation. The lessons that have emerged from the implementation of AFT can provide guidance for the development of prospective school-based vaccination programs, in particular for states that do not have HPV vaccine mandates.

## Figures and Tables

**Figure 1 vaccines-11-00946-f001:**
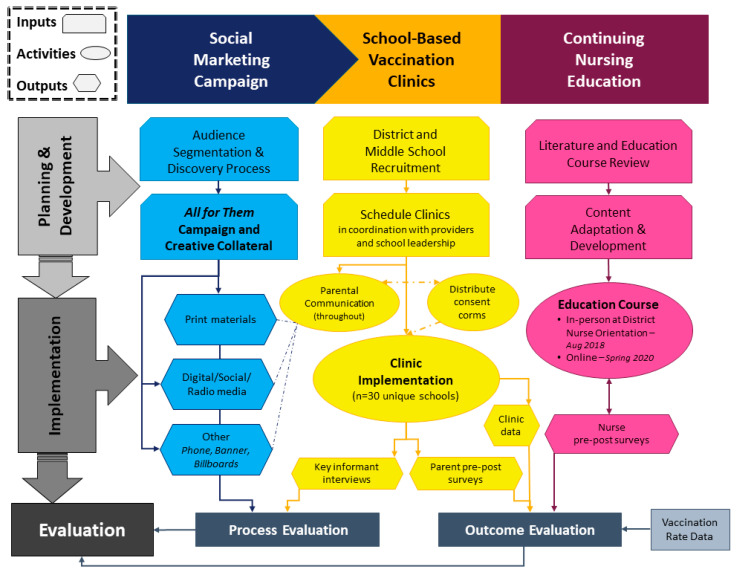
All for Them program development and implementation flowchart.

**Table 1 vaccines-11-00946-t001:** Domains, topics, and key lessons of All for Them.

Domains	Topics	Lessons
Strong champion	Collaboration since project inception Access to schools Parental communication School staff turnover	School principals were more likely to support and remain engaged; Parents were assured when it was clear that district management championed the program; Despite a high rate of staff turnover, having the district-level champion allowed new principals and nurses to continue with program implementation.
School-level support	School recruitment and retention Partnership with school staff School-level coordination District-level coordination Entry of recommended vaccines into district records	Working with a large, decentralized school district made recruitment and retention of schools a consistent challenge; Engaging with wraparound specialists, magnet coordinators, and Communities in Schools specialists increased access and communication to parents; Nurses can focus on tasks that only they can do if another coordinator is identified to perform other responsibilities; A designated bilingual nurse coordinator at the district level can support all nurses in participating schools and improve program quality; One-on-one training and encouragement for nurses to routinely enter all vaccines in a student’s immunization record are instrumental to reducing overvaccination, understanding HPV vaccination rates, and identifying opportunities for improvement.
Marketing	Response to evolving needs Monitoring efficacy Budget and resources	Marketing should be carefully planned out and tailored to individual schools and communities; Marketing strategies are to be modified as needed; It is beneficial to assess in-house skills and resources before approaching a marketing firm.
Mobile vaccination clinic collaboration	Provider partner selection Provider constraints	Mobile clinic organizations should be experienced and suitable for on-campus vaccination programs; Preferred schedules, staffing requirements, and potential issues of provider partners should be understood in advance to minimize disruption of the project.
Community presence	Trust and visibility Access to parents	Build trust and visibility through as many personal interactions with parents as possible; Informing parents of upcoming clinics, answering questions, and assisting them in completing paperwork can support maximum clinic participation.
Crisis management	Natural disasters COVID-19 pandemic	Be flexible and responsive to school and community needs and unforeseen circumstances, such as natural disasters or an infectious disease outbreak.

## Data Availability

Data sharing is not applicable to this article.

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
