# Peer review of "Lessons Learned from All for Them: Best Practices for a Cross-Collaboration Approach to HPV Vaccination in Public Schools"

_vaccines, 2023, doi:10.3390/vaccines11050946_

Round 1

Reviewer 1 Report

Paula M. Cuccaro et al. submitted to Vaccines an article, focusing to the best practices for a cross-collaborational approach to HPV vaccination in public school.

This topic is decidedly interesting, since Public Health practices, structured in a diversified way in different Countries, must guarantee appropriate vaccination strategies, offering the most appropriate healthcare service, included in the scope of a vaccination schedule. This manuscript is well structured and provides useful take-home messages, even if it is not a pure investigation, where statistical data are applied.

I would like to suggest to the Authors implementing this manuscript in the discussions, with the multi-professional aspects typical of these lines of work of the Public Health, providing concrete insights on the best vaccination practices applicable to your local context, with a focus on the healthcare workers involved in these practices (are there only nurses, since you are referring only to "school nurse continuing education", or other HCWs are involved?). In this regard, please evaluate the opportunity to discuss this topic, also focusing to the contents of the following works:

DOI: 10.3390/healthcare10101906

DOI: 10.3390/ijerph191710995

DOI: 10.1080/10810730.2021.1878312

DOI: 10.1080/21645515.2019.1565267

Thank you for your efforts in perfecting this article!

The English style used, both in terms of form and syntax, is adequate.

Author Response

We would like to thank the reviewers for their valuable insights and feedback. Below we have documented our response (in blue) to the reviewers’ comments as well as modified text (in red) for ease of viewing any changes made to the draft. We have also used the track changes feature in MS Word to easily mark the changes to the manuscript.

Reviewer #1

  1. I would like to suggest to the Authors implementing this manuscript in the discussions, with the multi-professional aspects typical of these lines of work of the Public Health, providing concrete insights on the best vaccination practices applicable to your local context, with a focus on the healthcare workers involved in these practices (are there only nurses, since you are referring only to "school nurse continuing education", or other HCWs are involved?). In this regard, please evaluate the opportunity to discuss this topic, also focusing to the contents of the following works:

Response: Thank you for the comment. We appreciate your recommendation of the literature regarding healthcare workers involved in vaccination practices, which has helped us realize that we were not entirely clear on the role of school nurses in our work. We would like to clarify that the school nurses were recipients of the intervention, supporting parents to increase HPV vaccine uptake of their adolescents, whereas other healthcare professionals external to the school system were part of the intervention to administer vaccines in mobile clinics. We have added the following in the revised manuscript (Lines 122-127):

Vaccines were administered by mobile clinic partners external to the school system. While school nurses in Texas do not routinely administer vaccines to public school students, this intervention provided education to increase their knowledge, positive attitudes, and effective communication with parents regarding the HPV vaccine. The goal was for school nurses to be confident in supporting parents’ decision-making and to make a strong recommendation for HPV vaccination.

Reviewer 2 Report

Overall, this is an extremely well written paper and lays out all of the necessary components of the program to be used as a model by others in this field. I commend the work being done by these researchers.

Abstract:

·         No comments – clearly written and all points are clear.

Introduction:

·         P.2, Line 50: Consider using non-Hispanic whites

·         Throughout the paper, make sure either US or U.S. is used consistently

·         P.2, Lines 94-102: There is no smooth transition into these sentences and they seem somewhat awkwardly placed. Perhaps separating out the information on stakeholders as a separate paragraph would help.

Materials and Methods:

·         Section 2.2: Please clarify if social media posts were only in English or other languages as well

·         Table 1 and P. 7, Lines 269-273: School-level support – Please clarify if nurses were also encouraged to enter vaccines in state immunization database.

Discussion:

·         No comments – extremely well written.

Author Response

We would like to thank the reviewers for their valuable insights and feedback. Below we have documented our response (in blue) to the reviewers’ comments as well as modified text (in red) for ease of viewing any changes made to the draft. We have also used the track changes feature in MS Word to easily mark the changes to the manuscript.

Reviewer #2

  1. P.2, Line 50: Consider using non-Hispanic whites

Response: Thank you for the comment. We have replaced whites with non-Hispanic whites.

  1. Throughout the paper, make sure either US or U.S. is used consistently

Response: Thank you for the comment. We have changed U.S. to US throughout the paper for consistency.

  1. P.2, Lines 94-102: There is no smooth transition into these sentences and they seem somewhat awkwardly placed. Perhaps separating out the information on stakeholders as a separate paragraph would help.

Response: Thank you for the comment. We have reordered the information on project stakeholders and placed it at the beginning of the program overview for clarity. The revised paragraph is as follows (Lines 88-95).

All for Them (AFT) was developed for implementation in a large urban public school district in Texas beginning in 2017 and began expanding in 2020 to include additional school districts varying in geographic region, size, and population. Project stakeholders included the partner school district(s), the participating middle schools, healthcare providers (i.e., mobile clinic partners), non-profit organizations focused on cancer prevention and immunizations, communication and content experts, and parents. Each of these stakeholder types was involved at various levels of the project, from conceptualization, development, and implementation to evaluation of the intervention.

  1. Section 2.2: Please clarify if social media posts were only in English or other languages as well

Response: Thank you for the comment. We have made clarifications that the social media advertisements and the scripted messages that schools used to promote vaccination clinics were bilingual, in English and Spanish (Lines 163-164; 172-173).

School-based support included provision of bilingual scripted messages that schools could use for different avenues of clinic promotion.

  1. Table 1 and P. 7, Lines 269-273: School-level support – Please clarify if nurses were also encouraged to enter vaccines in state immunization database.

Response: Thank you for the comment. We have made clarifications that school nurses are not authorized to enter vaccines in the statement immunization registry because they are not immunization providers of record. They were only encouraged to routinely enter all recommended vaccines into district records. The revision is as follows (Lines 286-289):

School nurses are not authorized to enter vaccines into the state immunization registry because they are not immunization providers of record. Therefore, we encouraged nurses to routinely enter all recommended vaccines, rather than only school-required vaccines, into district records.

Reviewer 3 Report

This is a very interesting paper about an innovative initiative for school-based HPV vaccination promotion. The combination of strategies makes a nice contribution and the findings from the interviews will inform public health practice. However, important details are missing from the report that detract from its utility. All of these are easily corrected, however.

 In particular we need more about:

(1) How were the schools recruited? The paper stresses the importance of support by a champion and at the school level but does not provide processes by which this was/can be accomplished.

(2) The text message campaign – What message strategies were used? It is not enough to provide the very general information provided. How is this sustainable without grants? What does it cost?

(3) How parental “pre-consent” was obtained? This is crucial to informing others.

(4) How the interviews were conducted and analyzed? What questions were asked? How long did they last? How were responses analyzed? What was the inter-coder reliability? Why weren’t parents and students interviewed?

 Also, I believe that the paper must note two important limitations. First, while a school-based approach is important it is probable impractical in certain school systems. Second, what are the barriers and obstacles to anticipate?

Author Response

We would like to thank the reviewers for their valuable insights and feedback. Below we have documented our response (in blue) to the reviewers’ comments as well as modified text (in red) for ease of viewing any changes made to the draft. We have also used the track changes feature in MS Word to easily mark the changes to the manuscript.

Reviewer #3

  1. How were the schools recruited? The paper stresses the importance of support by a champion and at the school level but does not provide processes by which this was/can be accomplished.

Response: Thank you for the comment. We have added the processes of recruiting schools and champions. We note that the district-level champion did not need to be recruited, as this person was actively involved in the development and implementation of the intervention since the inception of the project. Additionally, there was no formal recruitment of school-level champions; instead, school staff naturally filled in that role as needed depending on school needs, capacity, and personal enthusiasm for the project outcomes. The revisions are as follows (Lines 98-103, 209-212, 242-249):

Schools were eligible to participate if they had marked attendance boundaries in or adjacent to medically underserved areas and a majority racial and ethnic minority student population. Schools were recruited through individual informational meetings with principals and nurses and given the opportunity to commit to the project or not, depending on their capacity, and re-recruited as needed due to principal and nurse turnover.

Collaboration and coordination with a district-level champion (in this case, the district’s Director of Health and Medical Services) in the development and implementation since the inception of the program facilitated direct access to leadership at individual schools and effective communication to parents.

Given the unique needs and capacity of individual schools, school-level champions were identified on an ad-hoc basis by working with the principal and nurse to select individuals most appropriate to lead this effort at each campus. In some cases, where the nurse did not act as champion, school staff such as wraparound specialists, magnet coordinators, and Communities in Schools specialists filled that role. Individuals in these roles are usually in close contact with students, have trusted relationships with parents, and therefore served as excellent program advocates and de-facto champions, particularly if they exhibited personal enthusiasm for the project outcomes.

  1. The text message campaign – What message strategies were used? It is not enough to provide the very general information provided. How is this sustainable without grants? What does it cost?

Response: Thank you for the comment. We have specified the message strategies and tactics used for clinic promotion. We have also described that because we used the schools’ existing communication systems, there were no additional costs incurred by the project or the schools. We have added the following in the revised manuscript (Lines 172-178):

School-based support included provision of bilingual scripted messages that schools could use for different avenues of clinic promotion. Using the schools’ existing communication systems (e.g., robo-calls, text messages, social media channels) allowed the outreach to be conducted at no additional cost to the project or to the schools. Additional tactics included direct outreach by nurses to parents, nurse incentives, and a video for teachers with a project overview and instructions for supporting the nurse.

  1. How parental “pre-consent” was obtained? This is crucial to informing others.

Response: Thank you for the comment. We have further explained the process of obtaining consent from parents. The revision is as follows (Lines 118-123):

The project team delivered consent packets to participating middle schools, which were then distributed to all enrolled students to take home to parents. Interested parents completed and signed the forms, indicating the vaccines they consented for their child to receive. Students who returned completed forms to the school in advance received the vaccines needed on the scheduled clinic day. Parents were not required to be present during immunization. 

  1. How the interviews were conducted and analyzed? What questions were asked? How long did they last? How were responses analyzed? What was the inter-coder reliability? Why weren’t parents and students interviewed?

Response: Thank you for the comment. We would like to clarify that the interviews were conducted for the purpose of quality improvement and generating solutions to enhance the intervention. The results were not intended to be used for identifying significant patterns in the data by a formal qualitative analysis that includes traditional coding and calculation of inter-coder reliability. To bolster the key lessons learned, we conducted rapid assessment procedures to illustrate the key informants’ thoughts and suggestions relevant to the lessons. The interviews lasted from 30 minutes to an hour. Parents and students were not interviewed because the purpose of the interviews was to generate feedback from leaders at participating schools to enhance the program as it was being implemented. We have made revisions in Section 2.3 (Lines 184-195). 

We conducted semi-structured interviews as a quality improvement measure to generate feedback from leaders at implementing schools to enhance the program as it was being implemented. Interviews were conducted with school nurses, coordinators, and principals (n=21) who participated in AFT to understand their experiences with the program and what could be done to improve the process, including increasing parent and student participation. We conducted rapid assessment procedures to efficiently incorporate the key informants’ thoughts and suggestions as relevant evidence to the lessons learned. The interviews lasted from 30 minutes to an hour each. The interview guide included questions about priorities of their role as a school nurse/principal, perceived support for program implementation, planning and clinic logistics, perceived ownership of the program, parent engagement, and personal beliefs about vaccination.

  1. Also, I believe that the paper must note two important limitations. First, while a school-based approach is important it is probable impractical in certain school systems. Second, what are the barriers and obstacles to anticipate?

Response: Thank you for the comment. We acknowledge that school-based vaccination programs may not be feasible or suitable in certain school systems due to competing priorities, capacity for implementation, or availability of healthcare professionals to administer vaccines. We have also mentioned potential barriers and obstacles in each of the lessons learned along with solutions we implemented. We have added the following revision in the discussion (Lines 451-454):

However, we acknowledge that school-based vaccination programs may not be feasible or suitable in certain school systems due to competing priorities, capacity for implementation, or availability of healthcare professionals to administer vaccines.